# Research on Deformation and Failure Control Technology of a Gob-Side Roadway in Close Extra-Thick Coal Seams

**Bin Zhao** [1,2], **Shengquan He** [3,*], **Xueqiu He** [3], **Le Gao** [3], **Zhenlei Li** [3], **Dazhao Song** [3] and **Feng Shen** [3]

1    School of Safety Engineering, China University of Mining and Technology, Xuzhou 221116, China
2    China Coal Datong Energy Co., Ltd., Datong 037038, China
3    School of Civil and Resources Engineering, University of Science & Technology Beijing, Beijing 100083, China
*    Correspondence: shenqhe@163.com

**Abstract:** Close extra-thick coal seams are subject to the broken overburden of mined coal seams, and the deformation and damage of the roadways is serious, which affects the safe operation of the mine. To reduce the deformation and damage of the roadways, this paper studied the deformation and damage law of the gob-side roadway in close extra-thick coal seams through numerical simulation and field monitoring, compared and analyzed the deformation and damage characteristics of the roadway under different reinforcement support methods, determined the optimal reinforcement support method, and carried out field verification. The obtained results indicated that the deformation and damage of the gob-side roadway showed asymmetric characteristics. The large deformation of the coal body in the deep part of the roadway wall is an important reason for the continuous occurrence of roadway wall heave in the coal pillar. Under the action of unbalanced support pressure, the floor is subject to the coupling effect of horizontal extrusion pressure and vertical stress that cause extrusion mobility floor heave. The horizontal and vertical displacement of the coal pillar side of the roadway under different support methods is much larger than that of the solid coal side. Increasing the anchor cable length and fan-shaped arrangement can improve the support effect. Grouting at the coal pillar side can significantly improve the bearing capacity and stability of the coal pillar. The effect of floor grouting is much better than the anchor cable in controlling the floor heave. The integrated reinforcement method of anchor cable + coal pillar side grouting + floor grouting has the best effect with the least horizontal and vertical deformation. The research results are of great significance for ensuring the stability of similarly endowed roadways.

**Keywords:** close distance coal seams; roadway support; roadway deformation; grouting reinforcement; control technology

## 1. Introduction

In order to improve the mining rate of coal resources, reduce unnecessary mining lanes, and solve the problem of mining succession tension, gob-side entry retaining technology has been widely used [1–4]. Close extra-thick coal seams form a special overburden structure due to the mining of the overlying coal seams. Under the action of overburden structure, the deformation and damage of the gob-side roadway are serious. With the increase of mining depth, the degree of damage becomes more and more intense, causing damage to the roadway and equipment.

Related scholars have conducted a lot of research on the stress and deformation damage characteristics of the surrounding rock of gob-side roadways. Wang et al. [5] and Han et al. [6] investigated the structural mechanical causes of key block fracture at the roof of a gob-side roadway and found that bimodal stresses existed on both sides of the gob-side roadway and in the roadway in the low-stress area, and pointed out that the mechanical transfer mechanism of the surrounding rock in the vertical direction was weaker and the surrounding rock was most stable when the roadway was located outside the key block

fracture line. Li et al. [7] investigated the spatial and temporal evolution of fractures in the surrounding rock of a gob-side roadway under the pressure generated during mining, and the results showed that coal pillar deformation and rupture were the cause of destabilization and damage of the surrounding rock. Zheng et al. [8] established a mechanical model of the overburden rock of a gob-side roadway and analyzed the large deformation and stress field distribution of the roadway overburden rock, and the results showed that a large deformation of the roadway was caused by the stress field superimposed by the lateral transfer stress formed in the overburden rock formation, the oversupport pressure caused by mining, and the regional ground stress. Slashchov et al. [9] analyzed the minimum principal strain changes of rocks in different working faces and strata, and found that due to the existence of gob, the overburden pressure in part of the goaf decreased, but in the pressure bearing area of the solid coal pillar, the overburden of the main roof of the roadway was reactivated. Some scholars believe that filling the gob area can reduce the movement of the overburden rock layer and decrease the deformation of the gob-side roadway [10,11], but this method becomes more and more difficult to achieve with the increase of mining depth. Guo et al. [12] concluded that the length of the overhanging roof is the main factor affecting the stress distribution and deformation of the rock surrounding the roadway, and with the decrease of the overhanging length, the influence of the movement and load transfer of the overburden rock layer in the mining area on the stability of the tunnel. Liu et al. [13] analyzed the stress characteristics of the direct top of the gob-side roadway, and found that the roof deformation was symmetrically distributed, with the maximum roof deformation occurring in the middle of the roadway, and gradually decreasing with the increase of coal seam stiffness. Feng et al. [14] tested the mechanical properties of rock, and based on the energy analysis results, established a rock damage model by using the relative change of the dissipated energy ratio, and then evaluated the accuracy of the rock damage model with the peak strength and peak strain of rock as evaluation parameters.

Based on the research on the deformation mechanisms and deformation characteristics of the rock surrounding the gob-side roadway, the corresponding control measures are becoming more and more abundant. Studies have shown that the deformation and damage characteristics of roadways also have asymmetric characteristics due to the inhomogeneity of the stress conditions of the rock surrounding the roadways under the action of mining, so the control strategy based on asymmetric, high-prestressed anchor cables has a good effect on controlling the stability of the surrounding rock [15–17]. Peng et al. [18] studied the reasonable coal pillar size and roadway support method for isolated comprehensive mining workings in extra-thick coal seams using stress limit balance theory, the internal stress field distribution law, and the finite element method, and proposed the combined support method of "anchor mesh cable" + "reinforced ladder beam" + "steel belt", which can effectively reduce the displacement. Ren et al. [19] established a mechanical model and numerical simulation model to study the influence of the interaction between the bracket and the roof on the stability of the support of a gob-side roadway, and the results showed that the bracket resistance increased with the increase of width, and the optimal width was 3 m. Zheng et al. [20] derived a formula for calculating the maximum settlement of the roof slab and proposed the fenestration control technology of grating side-entry retaining, and the results showed that with the advancement of the working face, the deformation of the whole process of the gob-side roadway was effectively controlled. Xie et al. [21] and Xie et al. [22] analyzed the damage mechanism and controllable engineering factors causing deformation in the surrounding rock of the gob-side roadway, and proposed an asymmetric support scheme with prestressed anchor cable as the core member, which was verified in engineering practice. Based on the numerical simulation, Begalinov et al. [23] conducted a detailed study on the stress and deformation state of the rock around the transportation roadway, and adopted an adjustable support method according to the rock mass stability to control the deformation of the roadway to the maximum extent. He et al. [24] analyzed the deformation mechanism of the surrounding rock with theoretical and numerical simulations, and found that the high volume ratio of the inelastic zone is the

main reason for the severe deformation of the surrounding rock, and proposed a combined support technology with "reinforced anchor cable + W band + π beam + single hydraulic pillar" as the core, which was verified in the test section of the roadway. Yang et al. [25] concluded that coal pillars will gradually lose their bearing capacity under the influence of mining stress, leading to a sudden increase in the overhanging length of the roof and causing abnormal stress concentration in the tunnel surrounding rock, and proposed that strengthening the support density in the stress concentration area could effectively improve the surrounding rock deformation. Ma et al. [26] concluded that one of the reasons for the failure of the adit under the hard roof and soft floor is the insufficient support strength at the end of the gob-side roadway, so they proposed the support concept of "improving the support strength to achieve cutting the roof along the digging side", and the results showed that the anchor net, roof anchor, single hydraulic strut, and roof bolting technology can ensure the support effect. Guo et al. [27] found that after the damage of the main roof in the dynamic pressure zone, its rotary sinking sharply increased the deformation and damage of the tunnel envelope, and proposed a support design consisting of high tension, constant resistance, and the use of large deformation anchors rods + single hydraulic strut + steel beam + U beam. Li et al. [28] studied the mechanical properties of grouted rock from both macroscopic and microscopic aspects by using shear and compression experiments, and found that the elastic modulus of rock mass was greatly improved after grouting. MałKowski et al. [29] studied three different roadway support schemes for six years, and determined the time of secondary equilibrium and the strain of support unit after the rock mass is damaged so as to guide the rationality of support installation under specific stope and geological conditions. Dychkovsky et al. [30] studied the visualization of the formation principle of the stress–strain state in the coal support pressure area and evaluated the rock mass occurrence conditions. The results showed that when the width of the support pressure area was 18 m, the maximum stress was reduced to 70 MPa, which can be used as the basis for identifying the bearing capacity of the mine and the excavation roadway.

In summary, it can be seen that fruitful results have been achieved for the deformation and damage characteristics and support technology of gob-side roadways, but not much research has been conducted for the gob-side roadway support of the overburden coal seams that have been mined under the condition of close extra-thick coal seams. However, in this occurrence condition, the deformation and damage of the roadway are serious; the stability is low, and it is easy to cause damage to personnel, property, and equipment. The existing research results can not meet the requirements of roadway deformation control. Therefore, it is necessary to apply further efforts to improve the stability of the roadway. In this paper, we studied the deformation and damage law of a close extra-thick coal seam gob-side roadway through numerical simulation, compared and analyzed the deformation and damage characteristics of the roadway under different reinforcement support methods, determined the optimal reinforcement support method, and carried out field verification.

## 2. Background and Method

### 2.1. Engineering Overview

The 305# coal seam of the Tashan coal mine is one of the few extra-thick coal seams in China, with an average burial depth of 435 m and a variation coefficient of 0.16%, and which is a stable coal seam. The 4.67 m layer above the coal seam 305# is the 102# coal seam of the Carboniferous Taiyuan Group, with an average seam thickness of 3 m, and mining was completed in July 2015. The protected coal pillar between the 10,201 working face and 10,203 working face mining void area of coal seam 102# is 20 m, as shown in Figure 1a. At present, the mining of the 30,503 working face of coal seam 305# has been started by using long-wall comprehensive mining to release the top coal-recovery process, which is typical in mining of extra-thick coal seams under close mining areas, and its gob-side 30,501 working face has been completed. The 30,503 working face has a strike length of 1738.58 m and a tendency length of 193 m. The average thickness of coal seam 305# is 14 m, and between it and coal seam 102# is 4.67 m-thick mudstone seam.

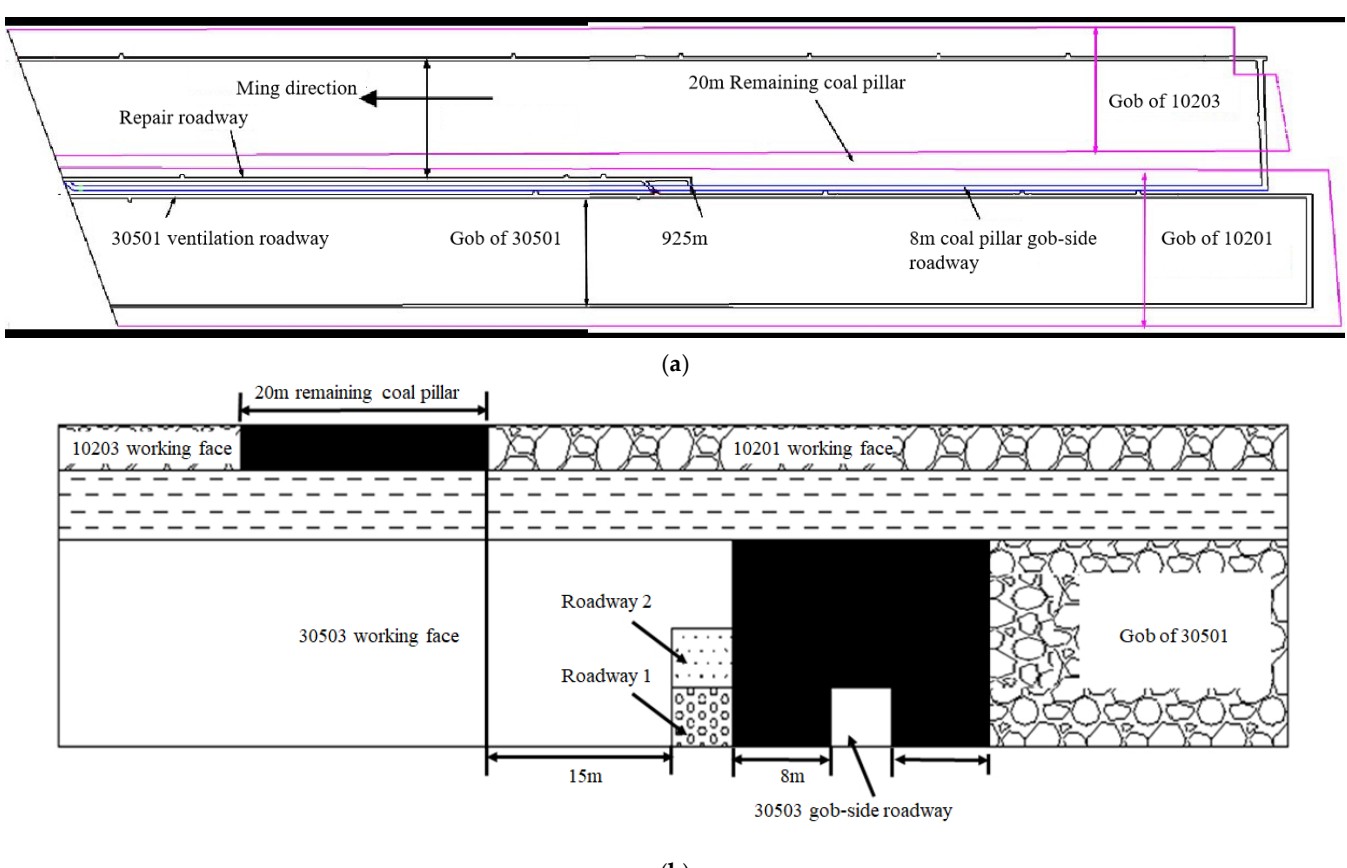

**Figure 1.** Diagram of the 30,503 working face layout. (**a**) Layout plan. (**b**) Cutaway layout.

The initial design of the 30,501 working face used roadway 1 as the transportation roadway; the horizontal distance of this roadway from the coal pillar left in coal seam 2# was 15 m. When the roadway was dug to 925 m, due to the serious mine pressure, the roadway dig program was changed and roadway 2 was used instead. The current mining is underway for the 30,503 working face of the coal seam 305# seam of the Carboniferous Taiyuan Group. It is planned to repair roadway 3 above roadway 1 as the return airway of 30,503, and in the process of repairing to 925 m, due to the structural damage of roadway 1 and serious roof fall, the working face of 30,503 was rearranged by the method of leaving an 8 m coal pillar in the gob-side entry driving, and the repair roadway was filled with high water material. After the roadway layout of working face 30,503, the horizontal distance between the roadway and the remaining coal pillar of overlying coal seam 102 # is 28 m. The gob-side is a rectangular section along the floor with a height of 4.0 m and a width of 5 m. Under the disturbance of back mining at the working face, the deformation and damage of the gob-side roadway is intense when crossing the filling roadway, which seriously affects the safe back mining at the working face, and the roadway layout is shown in Figure 1. The gob-side roadway is supported by a combination of an anchor rod and an anchor cable. The roadway is supported by left-handed, high-strength, rebarless anchor rods with a roof anchor rod size of Ø22 mm × 2400 mm and a side anchor rods size of Ø20 mm × 2200 mm with a preload torque of no less than 400 N·m. The anchor cable adopts 1 × 19 strands of Ø22 mm × 8300 mm high-strength low-relaxation prestressing strand with an ultimate breaking force of ≥550 kN and a pretensioning force of 200 kN. The roof anchor rods are 6 rows each with inter-row spacing of 1000 mm × 1000 mm, the side anchor rods are 4 rows each with inter-row spacing of 1000 mm × 1000 mm, and the 5 anchor cables are symmetrically arranged along the centerline and both sides of the roadway with inter-row spacing of 1300 mm × 2000 mm as shown in Figure 2. Affected by the mining stress of the working face and the complex mining environment around the gob-side roadway, the

deformation and damage of the gob-side roadway under the existing support conditions is still serious. The middle and upper part of the roadway are greater than the lower part, and the coal pillar side is greater than the solid coal side, as shown in Figure 3.

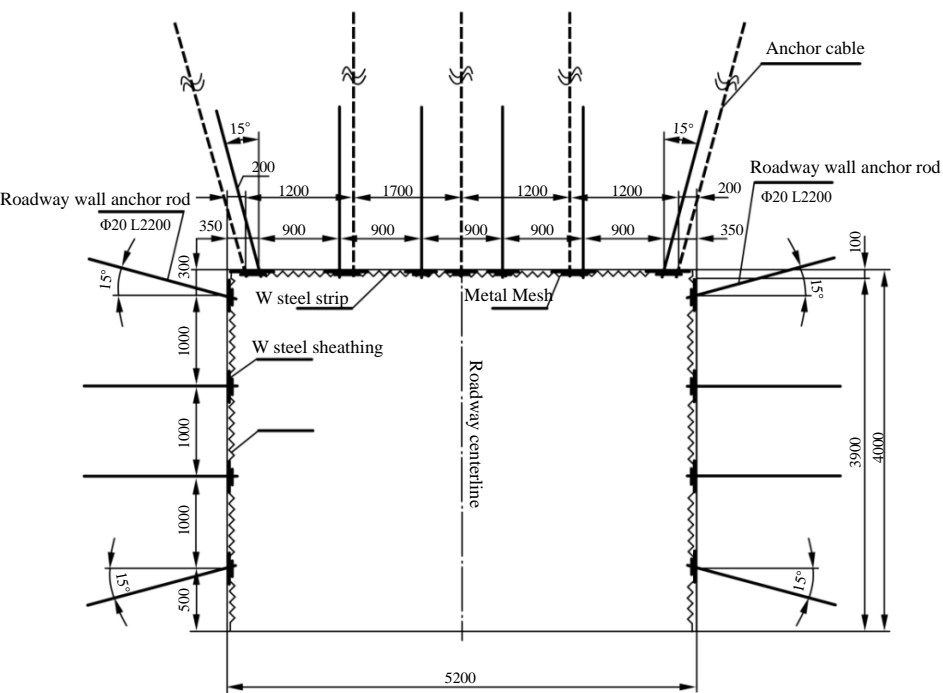

**Figure 2.** Schematic diagram of section support of the 30,503 gob-side roadway.

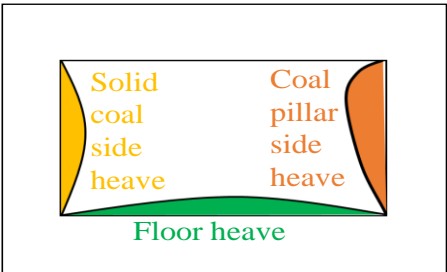

**Figure 3.** Deformation and damage characteristics of the roadway.

*2.2. Model Construction*

The FLAC3D numerical model was established with the geological background of the Tashan coal mine, as shown in Figure 4. The coal seam 305# is buried at 435 m, and the horizontal elevation is between +1005~+1019 m. Considering the influence of repeated mining of the coal seam, the model simulated back mining for both coal seams 102# and 305#. Considering the size of the model and the computational efficiency, the top burial depth of the model was set to 335 m. The model size is 640 m × 400 m × 130 m (XYZ) with 530,504 grids. The thickness of coal seam 305# in the model was 14 m, the thickness of immediate roof was 4.6 m, the thickness of coal seam 102# was 3 m, and the width of the remaining coal pillar was 20 m. The width of the 30,503 gob-side roadway was 5 m, the height was 4 m, the west part of the gob-side roadway was 8 m solid coal area for roadway 1 and 3 areas, the east part was 8 m narrow coal pillar for the mining area, the west part was 15–35 m solid coal for the overburden of coal seam 102#'s left coal pillar area, and the rest of the overburden rock thickness is given according to the actual situation of the mine.

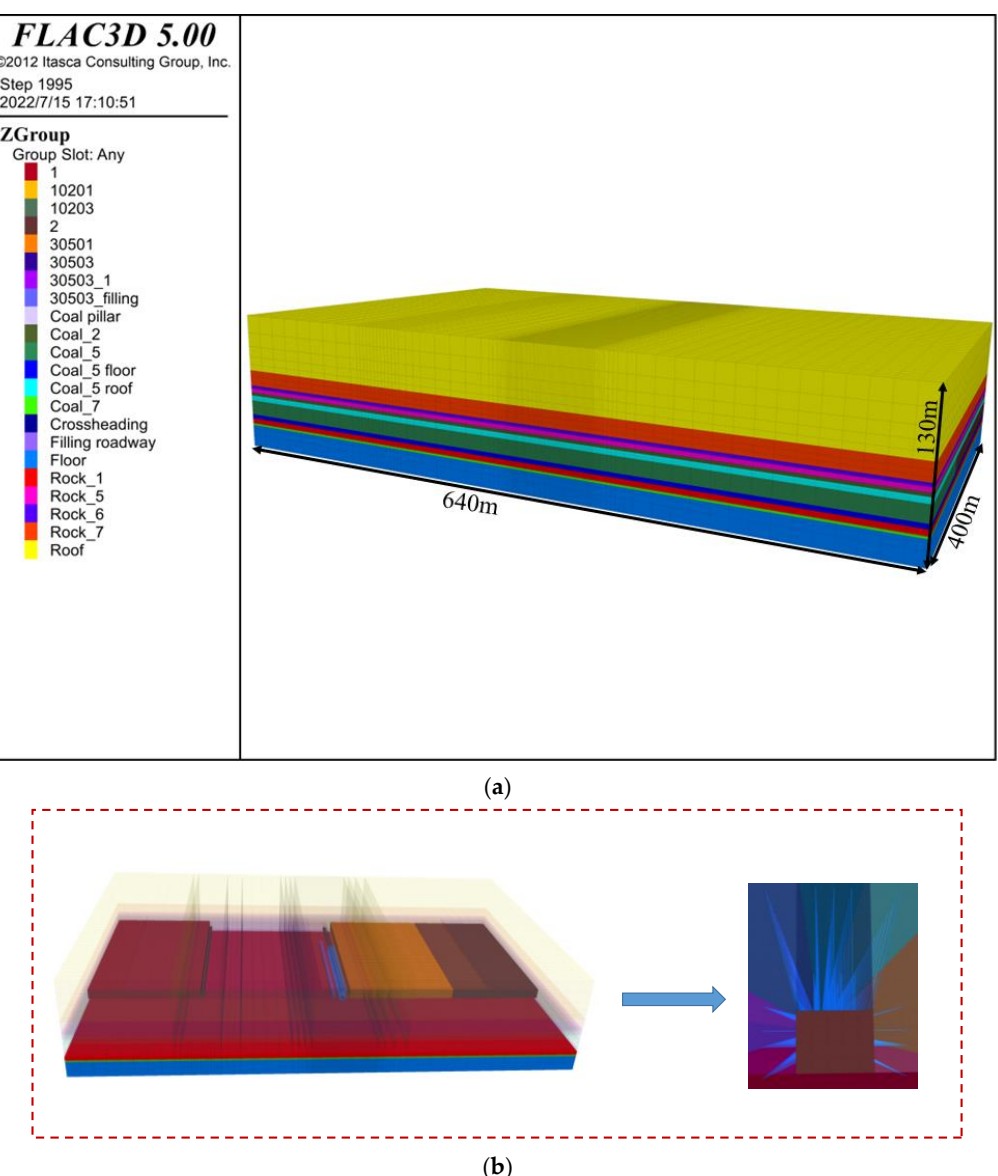

(**a**)

(**b**)

**Figure 4.** Numerical model. (**a**) Numerical model. (**b**) Model cross-section.

This numerical study selected the Mohr–Coulomb intrinsic structure model to study the deformation and damage control technology of a gob-side roadway in close extra-thick coal seam, and the physical and mechanical parameters used in the numerical model were obtained from the laboratory experimental results of coal rock samples in Tashan mine, as shown in Table 1. The cohesion and tensile strength were measured. The bulk modulus, shear modulus, and internal friction angle were calculated as follows:

$$K = \frac{E}{3(1-2\mu)} \tag{1}$$

$$G = \frac{E}{2(1+\mu)} \tag{2}$$

$$\varphi = 30\left(1 - \frac{\mu}{1-\mu}\right) + 15 \tag{3}$$

**Table 1.** Physical and mechanical parameters of the coal rocks.

| Rockiness | Thickness (m) | Density (kg/m³) | Bulk Modulus (GPa) | Shear Modulus (GPa) | Cohesion (MPa) | Internal Friction Angle (°) | Tensile Strength (MPa) |
|---|---|---|---|---|---|---|---|
| Sandy mudstone | 51 | 2400 | 5.08 | 3.5 | 2.78 | 32.21 | 1.32 |
| Clasticite | 15 | 2450 | 5.49 | 3.78 | 2.94 | 33.15 | 1.52 |
| Fine Sandstone | 3 | 2470 | 8.77 | 6.58 | 4.77 | 36.79 | 2.98 |
| Sandy mudstone | 8 | 2400 | 5.08 | 3.5 | 2.78 | 32.21 | 1.32 |
| 2# Coal | 3 | 1340 | 4.93 | 3.25 | 2.67 | 31.22 | 1.04 |
| Kaolinite | 5 | 2500 | 7.8 | 7.63 | 5.86 | 32.17 | 1.57 |
| 3~5# coal | 14 | 1340 | 4.93 | 3.25 | 2.67 | 31.22 | 1.04 |
| Kaolinite mud | 4 | 2546 | 6.65 | 4.33 | 3.63 | 35.83 | 2.38 |
| Siltstone | 5 | 2470 | 7.81 | 5.62 | 4.24 | 35.93 | 2.71 |
| 7# Coal | 2 | 1340 | 4.93 | 3.25 | 2.67 | 31.22 | 1.04 |
| Sandy mudstone | 21 | 2400 | 5.08 | 3.5 | 2.78 | 32.21 | 1.32 |
| High water materials | 4 | 1310 | 1.1 | 0.7 | 0.25 | 31.22 | 0.5 |

The parameter values of the abutment structure elements are shown in Table 2.

**Table 2.** Parameter values of the abutment structure elements.

| Type | Bolt Length/mm | Grout Length/mm | Diameter/mm | Tensile Strength/kN | Row Spacing/mm |
|---|---|---|---|---|---|
| Anchor rods | 8300 | 2700 | 20 | 200 | 1300 × 1000 |
| Anchor cables | 2400 | 800 | 22 | 120 | 1000 × 1000 |

### 2.3. Boundary Condition

The bottom of the model was set as a fixed boundary, the stress boundary conditions were applied around the model, the boundary conditions of the model were set by the results of ground stress measurements, the model vertical load was applied according to the change of burial depth, and the horizontal stress was calculated on the left and right according to the lateral pressure coefficient of this coal bed. Field measurements yielded a vertical load size of 8.25 MPa at the top of the model and a lateral pressure coefficient of 1.2, and then the horizontal stress at different burial depths along the vertical could be calculated by the following equation:

$$\sigma_{xx} = 1.2\gamma H \tag{4}$$

$$\sigma_{yy} = 1.2\gamma H \tag{5}$$

where $\gamma$ is unit weight, $H$ is the depth below the ground surface.

### 2.4. Roadway Monitoring Line Layout

Numerical simulations showed that 20 m ahead of the working face is the peak area of mining stress. Therefore, in order to quantitatively reveal the deformation and damage characteristics of the two sides and the floor of the gob-side roadway under the existing support conditions, eight monitoring lines were arranged at the position of 20 m ahead of the working face as shown in Figure 5. The monitoring line A is located at the height of the roof of the roadway. With an interval of 1 m, a total of 4 monitoring lines were arranged. The floor monitoring line starts from the position of 1 m from the floor, with an interval of 1 m, and a total of 4 lines were arranged. The section size of the roadway was 5 m × 4 m. The length of the monitoring line at the solid coal side and the coal pillar side was 8 m, and the length of the floor monitoring line was 6 m. All monitoring lines were arranged parallel to the roadway floor.

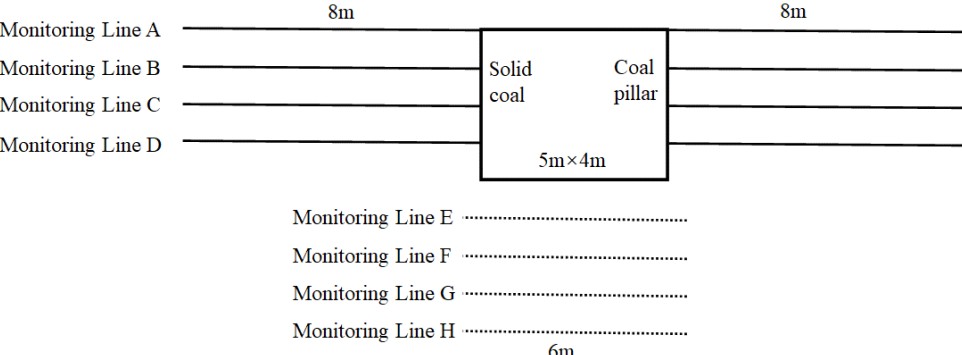

**Figure 5.** The 30,503 gob-side roadway monitoring line layout.

## 3. Results and Discussion

### 3.1. Deformation and Damage Characteristics of the Gob-Side Roadway under Existing Support Conditions

When the working face is mined to 100 m in a direction, the horizontal displacement distribution laws of each monitoring line of the two sides of the roadway 20 m ahead of the working face are shown in Figure 6. Due to the roof sinking, the roof of the roadway is no longer at the height of the monitoring line A, so the data of monitoring line A are not analyzed. It can be seen from Figure 6 that under the influence of mining, the horizontal displacements of the monitoring lines on two sides of the roadway were basically the same. The maximum deformation of the solid coal side was 0.69 m, which occurred 1 m from the roadway wall and then gradually decreased. The horizontal displacement of coal pillar side near the 30,501 gob area showed a continuous increasing trend, and the maximum displacement of the coal pillar side could reach 1.4 m. It can be seen by comparing the displacement of the coal wall of the coal pillar that the deformation at B and C monitoring lines in the middle and upper parts of the roadway were larger than that at D, showing asymmetric deformation characteristics, consistent with the site deformation characteristics. Based on this, it can be seen that the large deformation of the coal body in the deep part is an important reason for the continuous occurrence of roadway wall heave in the coal pillar, and the roadway wall heave in coal pillars cannot be controlled by using only anchor rods and anchor cables for reinforcement support. The deformation within 2~3 m of the solid coal side within the influence area of mining is still large, and the existing 2.2 m anchor rods support method of the roadway side cannot control the roadway wall heave. Improving the bearing capacity of the coal pillar side and using longer anchor cables to support the two sides are feasible measures to improve the stability of the gob-side roadway.

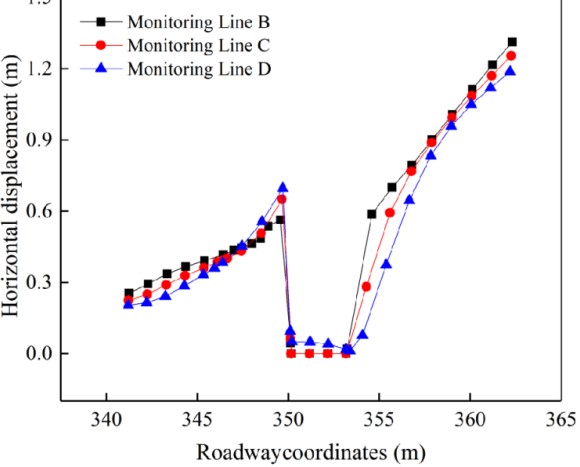

**Figure 6.** The horizontal displacement distribution law of the two sides of the gob-side roadway at 20 m ahead of the working face.

The vertical displacement of each monitoring line at different distances from the floor is shown in Figure 7. The floor of the roadway was obviously affected by mining. The maximum floor heave of each monitoring line basically occurred in the middle of the roadway, and the maximum displacement at 1 m from the roadway floor was 0.41 m. Under the action of high stress in the two sides of the roadway, the floor heave occurred continuously, and the displacement decreased gradually with the increase of the distance from the floor. The displacement was almost zero at the position of 4 m from the floor, less affected by mining and static load stress. For the 30,503 gob-side roadway, the vertical stress generated after the mining-induced overburden high-level and low-level main roof breakage acted on the gob-side rock surrounding the roadway, and the concentrated stress was transmitted to the floor through the two sides of the roadway. The floor was extruded into the roadway by the coupling of horizontal extrusion pressure and vertical stresses under the action of unbalanced support pressure, and the extruded fluid floor heave occurred [31].

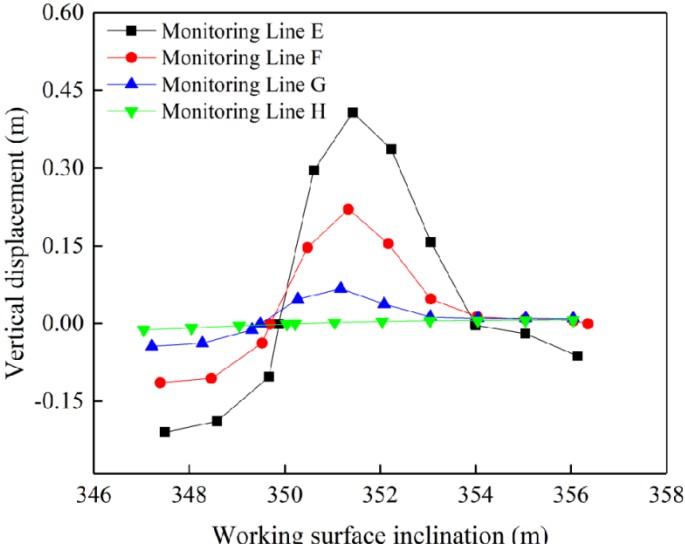

**Figure 7.** The vertical displacement distribution of the floor 20 m ahead of the working face (the monitoring line layout map shows 6 m).

### 3.2. Stress and Displacement Distribution Law of the Roadway Wall under the Roadway Wall Reinforcement

3.2.1. Optimization of the Support Method of the Roadway Wall

Based on the deformation and damage characteristics of the 30,503 gob-side roadway, it is known that the existing anchor rod support depths are shallow and do not achieve effective support, and the coal deformation inside the coal pillar side of the roadway is large. The support of the two sides under the existing support method of the roadway should be optimized to ensure the safety of the subsequent working face retrieval. Therefore, long anchor rods or anchor cables and coal pillar side grouting were considered for reinforcement support. Considering the cost of support, the existing support materials of the mine were used for reinforcement support, including 2.2 m and 2.4 m anchor rods and 8.3 m anchor cables. With the existing support method, the length of the two sides anchor rods is 2.2 m and the spacing is 1.0 m. The effective control of the roadway can no longer be achieved by using the existing anchor rods, and the number of anchor rods may aggravate the shallow coal fragmentation by reinforcement, which further leads to the increase of deformation. Therefore, the existing 8.3 m anchor cables were divided into two 4.15 m-long anchor cables for reinforcement support of the two sides. Under the influence of mining, the middle part of the solid coal side heaves, and the middle and upper part of the coal pillar side heaves seriously. Therefore, the roadway sides reinforcement method was carried out for these parts, and the anchor cable reinforcement design includes the following four options: (1) three 4.15 m anchor cables were used to support the two sides, which were arranged

in the middle of the existing two rows of anchor rods and arranged in 212 three flowers with the existing anchor rods. The distance between rows was 1.0 m, 2.0 m, the first row was 0.8 m from the top of the roadway wall, and the third row was 1.2 m from the floor, as shown in Figure 8a. (2) Three anchor cables were used; the inter-row spacing was 1.3 m, 2.0 m. The first and third rows were 0.8 m from the roof and floor, they were reinforced at 15° to the roadway wall, and the middle anchor cable was perpendicular to the roadway wall, as shown in Figure 8b. (3) Four anchor cables were used, with inter-row distance of 1.1 m and 2.0 m. The first row was 0.3 m from the roof and 15° from the roadway wall and the remaining three rows were arranged vertically on the roadway wall, with inter-row distances of 1.2 m and 2.0 m, as shown in Figure 8c. (4) Four anchor cables were used, with inter-row spacing of 1.2 m, 2.0 m. The first row was 0.3 m from the roof and 15° from the roadway wall, the second and third rows were perpendicular to the roadway wall, and the fourth row was 0.9 m from the floor and 15° from the roadway wall, as shown in Figure 8d. Different support schemes have different support effects on the roadway due to the different number of anchor cables, the row spacing between anchor bolts, and the angle between the anchor cables and the two sides. Under certain conditions, the greater the support density, the longer the support length, and the more the support angle conforms to the stress distribution, the more obvious the support effect that can be achieved.

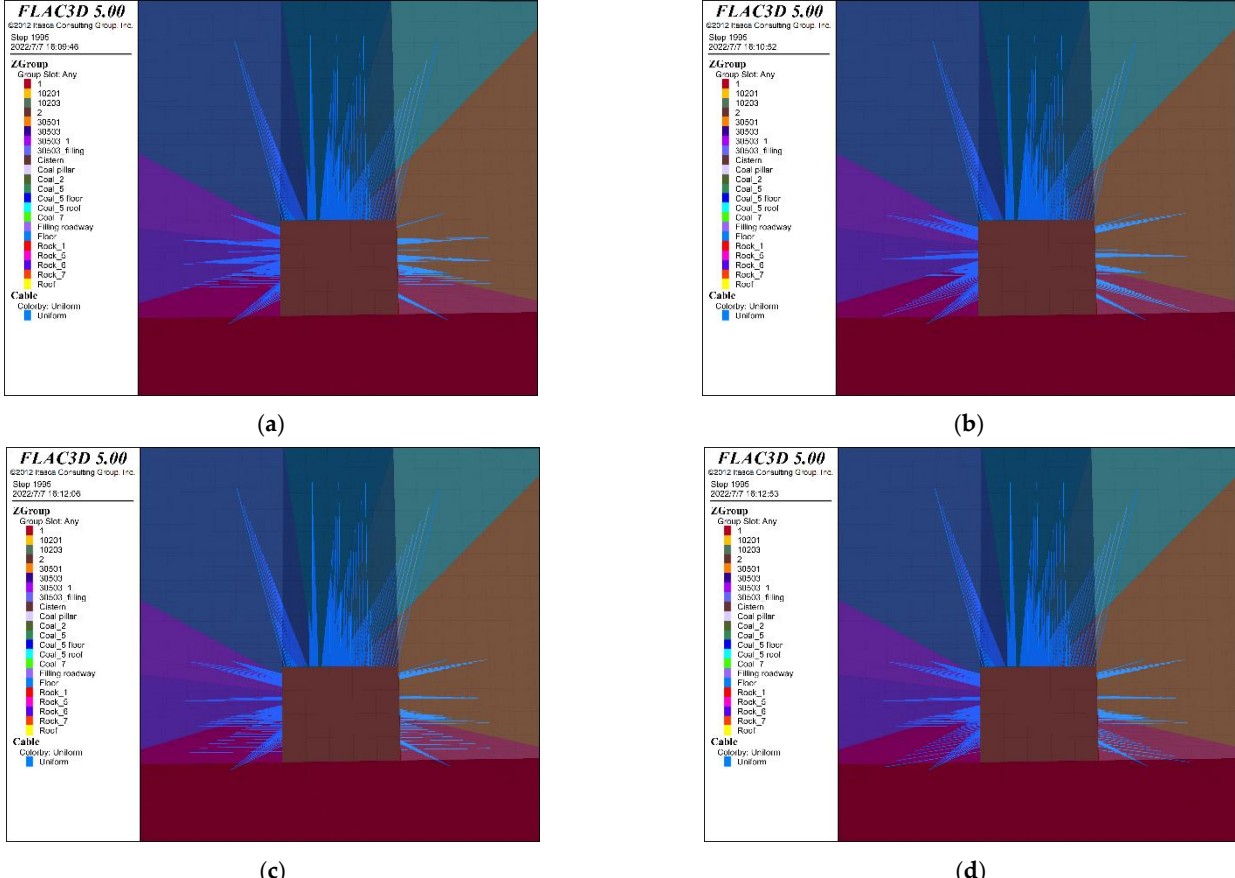

**Figure 8.** Anchor cable reinforcement method for the two sides of the roadway. (**a**) Reinforcement support method 1. (**b**) Reinforcement support method 2. (**c**) Reinforcement support method 3. (**d**) Reinforcement support method 4.

Based on the results of on-site drilling and numerical simulation analysis, the broken coal pillar is an important cause of wall heave, so grouting reinforcement support was carried out on the side of the coal pillar. The reinforcement method mainly includes four methods with the grouting depths of 2 m, 3 m, 4 m, and 5 m at the side of the coal pillar. The research shows that when the coal is grouted, the stiffness and shear strength of the

fracture surface of the coal and rock mass will be greatly improved, thus improving the strength and bearing capacity of the coal and rock mass [32–34]. In this paper, we simulated the effect after grouting by increasing the strength and stiffness of the coal body within the grouting range when simulating the grouting support on the coal pillar side. In the model, the strength and stiffness and friction of the coal body increased by 1.2 times compared with the parameters in Table 1. The diagram of grouting range is shown in Figure 9.

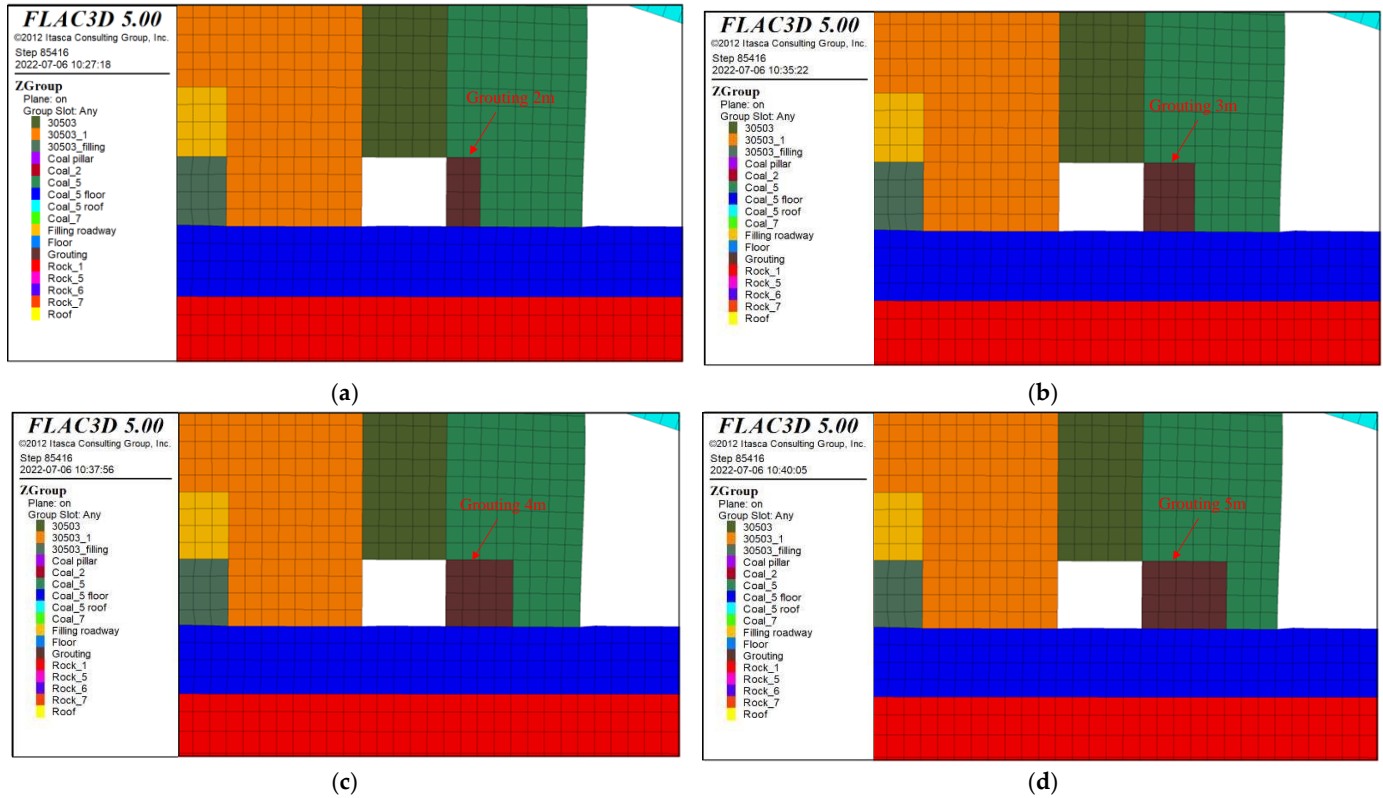

**Figure 9.** Coal pillar side grouting reinforcement method. (**a**) Grouting 2 m. (**b**) Grouting 3 m. (**c**) Grouting 4 m. (**d**) Grouting 5 m.

### 3.2.2. Deformation and Damage Characteristics of the Roadway Wall

When the working face was mined to 100 m strike, the horizontal displacement distribution law of each monitoring line on the two sides of the roadway 20 m ahead of the working face under different reinforcement support methods is shown in Figure 10. The horizontal displacement variation law of each monitoring line under different support methods was basically the same, and the horizontal displacement of the roadway wall on the coal pillar side was much greater than that on the solid coal side. Compared with the horizontal deformation of the two sides of the gob-side roadway under the original support, each reinforcement support method reduced the deformation of the two sides of the roadway and played a role in enhancing the support capacity. Among them, the first reinforcement method had the worst effect, the second reinforcement and third reinforcement had few differences, and the fourth reinforcement method had the best control effect. Additionally, the effect of the more broken coal pillar side was more obvious under the anchor cable-reinforced support, compared with the original support deformation reduced by more than 20%, as shown in Tables 3 and 4.

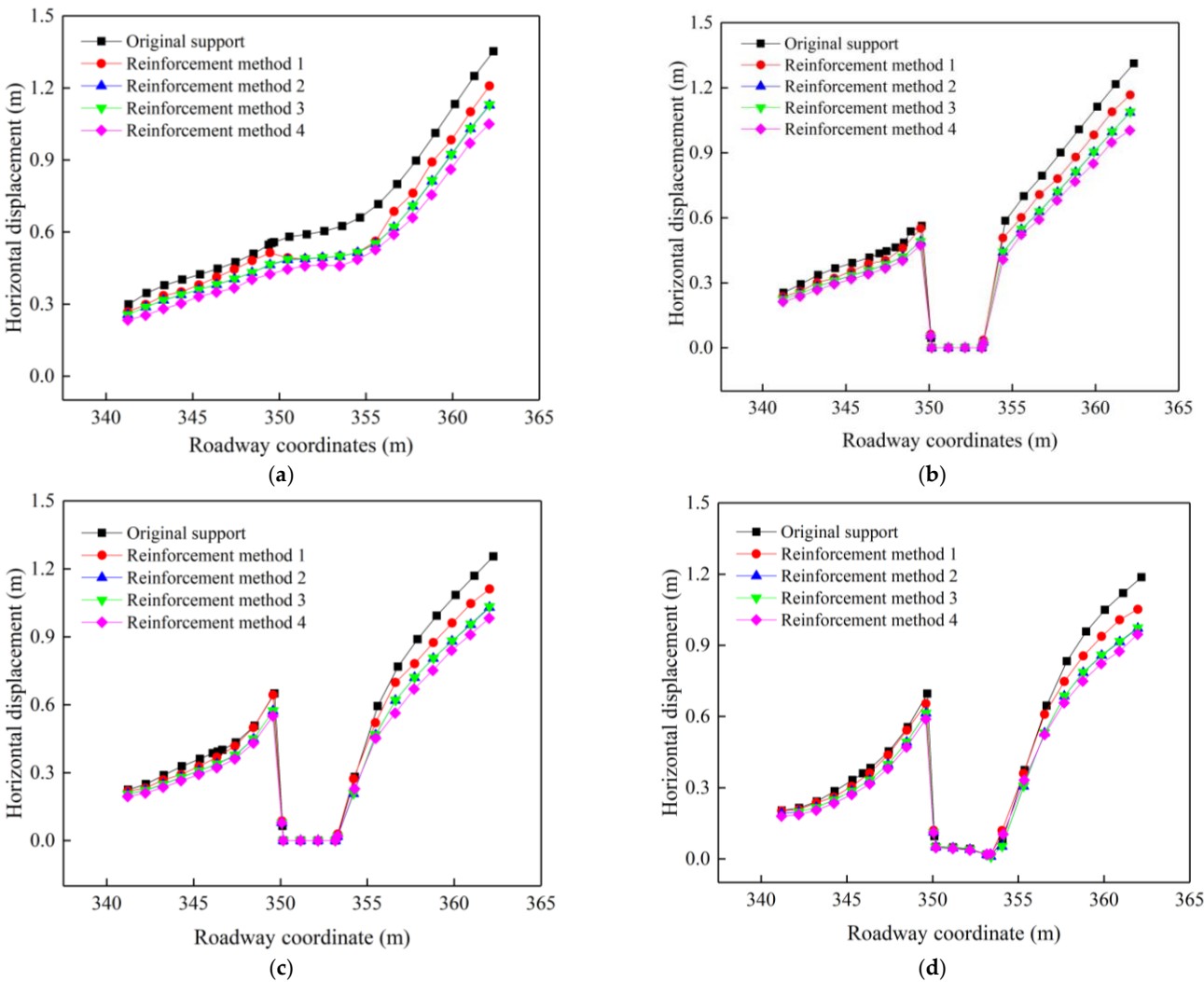

**Figure 10.** Horizontal displacement of the two sides of the roadway under the anchor cable reinforcement method. (**a**) Monitoring Line A. (**b**) Monitoring Line B. (**c**) Monitoring Line C. (**d**) Monitoring Line D.

**Table 3.** Percentage reduction in deformation under solid coal side anchor cable reinforcement.

| Reinforcement Method/ Location | Method 1 | Method 2 | Method 3 | Method 4 |
|---|---|---|---|---|
| Monitoring Line A | 5.00% | 14.80% | 14.80% | 22.00% |
| Monitoring Line B | 5.00% | 11.00% | 11.00% | 16.00% |
| Monitoring Line C | 11.00% | 14.00% | 14.00% | 18.70% |
| Monitoring Line D | 5.00% | 16.00% | 16.00% | 20.00% |

**Table 4.** Percentage reduction of deformation under coal pillar side anchor cable reinforcement.

| Reinforcement Method/ Location | Method 1 | Method 2 | Method 3 | Method 4 |
|---|---|---|---|---|
| Monitoring Line A | 11.80% | 19.80% | 19.8% | 27.70% |
| Monitoring Line B | 15.50% | 22.20% | 22.2% | 27.00% |
| Monitoring Line C | 11.30% | 18.20% | 18.2% | 26.00% |
| Monitoring Line D | 10.80% | 18.00% | 18.0% | 22.00% |

The vertical displacement distribution on the two sides of the roadway under different anchor cable reinforcement methods is shown in Figure 11. It can be seen from Figure 11

that the vertical displacement control on the two sides of the roadway was obvious with different reinforcement methods, especially on the coal pillar side, which indicates that the use of anchor cable reinforcement at this stage can improve the stability of two sides. Comparing the four reinforcement methods, we can see that the fourth reinforcement method had the best effect, followed by the second and third, and the first one had the worst effect. In order to determine the reinforcement effect of anchor cable reinforcement support on the shallow part of solid coal and the deep part of the coal pillar, the deformation at 1 m in the shallow part of the solid coal side and 4 m in the middle of coal pillar side under different reinforcement methods were compared with the deformation under the original support method, and the results are shown in Tables 5 and 6. The deformation of the coal pillar side's vertical pressure under the influence of mining was more serious, the effect of using reinforcement anchor cable was obvious, the first one could reduce the deformation by about 10%, and the best effect was achieved by using the fourth reinforcement method, which could reduce the deformation by more than 20% compared with the original one.

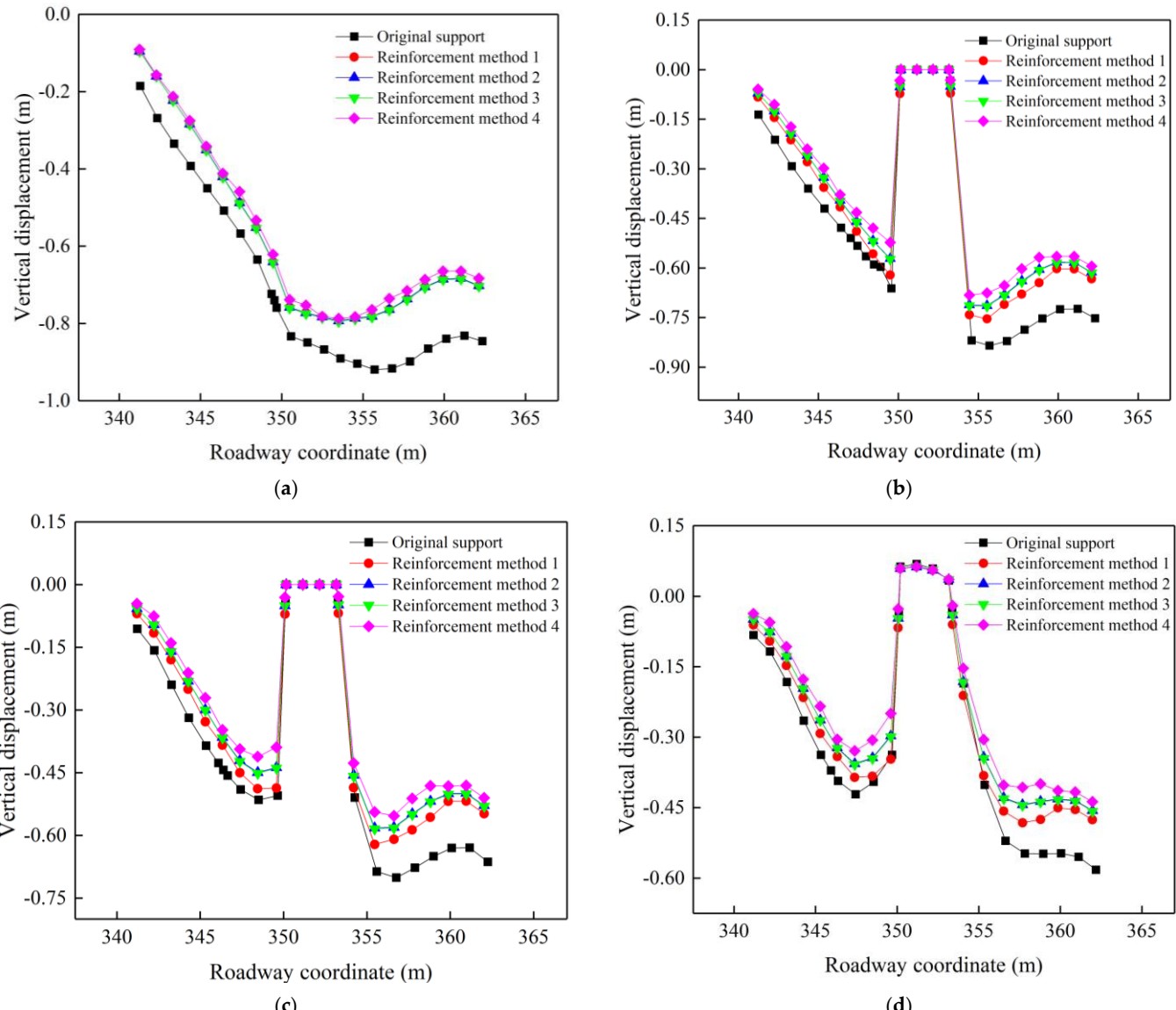

**Figure 11.** Vertical displacement of the two sides of the roadway under the anchor cable reinforcement method. (**a**) Monitoring Line A. (**b**) Monitoring Line B. (**c**) Monitoring Line C. (**d**) Monitoring Line D.

**Table 5.** Percentage reduction in deformation under solid coal side anchor cable reinforcement.

| Reinforcement Method/Location | Method 1 | Method 2 | Method 3 | Method 4 |
|---|---|---|---|---|
| Monitoring Line A | 14.00% | 15.60% | 15.60% | 17.00% |
| Monitoring Line B | 6.00% | 13.00% | 13.00% | 21.00% |
| Monitoring Line C | 4.00% | 11.70% | 11.70% | 27.50% |
| Monitoring Line D | 4.800% | 12.20% | 12.20% | 26.00% |

**Table 6.** Percentage reduction of deformation under coal pillar side anchor cable reinforcement.

| Reinforcement Method/Location | Method 1 | Method 2 | Method 3 | Method 4 |
|---|---|---|---|---|
| Monitoring Line A | 10.00% | 20.00% | 20.00 | 21.30% |
| Monitoring Line B | 13.00% | 17.90% | 17.90% | 23.00% |
| Monitoring Line C | 12.00% | 19.40% | 19.40% | 25.00% |
| Monitoring Line D | 14.20% | 21.40% | 21.40% | 28.00% |

All reinforcement methods play a role in reducing the deformation of two sides of the roadway, mainly because the 4.15 m long anchor cable can stabilize the broken coal body in the deeper part of the sides compared with the existing 2.2 m-long anchor rods, improving the range of the stability area of the roadway side. The main reason why method 4 is more effective than other support methods is that its fan-shaped arrangement provides multi-angle support to two sides of the roadway and covers a wider area than the vertical roadway.

The coal pillar side is severely broken due to multiple mining disturbances. In order to determine the reasonable grouting depth of the coal pillar side of the roadway wall, vertical stress data were extracted from each monitoring line position of the two sides of the roadway under different grouting depths, as shown in Figure 12. The vertical stress distribution on the solid coal side was basically unchanged compared to the original support method, and the peak stress on the coal pillar side was significantly increased compared with that without grouting. Additionally, this shows that grouting the coal pillar can effectively improve the bearing capacity and stability of the coal pillar. Comparing the stress distribution on the side of the coal pillar when the grouting depth was 2 m, 3 m, 4 m, and 5 m, we can see that the peak stress still appeared at 2 m. It shows that the low-level main roof broke at the position 2 m above the coal pillar, resulting in the formation of a stress concentration zone on the coal pillar. When 2 m grouting was used, the peak stress at each monitoring point on the coal pillar basically reached 23~26 MPa. With the increase of grouting depth, the stress distribution gradually changed to bimodal distribution, with the peak stress near the roadway wall gradually decreasing and the peak stress near the gob area gradually increasing, indicating that the integrity of the coal pillar is continuously improving. The vertical stress distribution in the ungrouted area on the side of the gob is basically the same as under the original support method.

To further determine the reasonable depth of coal pillar side grouting, the horizontal deformation of two sides after different grouting depths and anchor cable support are shown in Figure 13. Although only the shallow part of the coal pillar side was grouted, it still could significantly reduce the deformation of two sides of the roadway, indicating that improving the bearing capacity and integrity of the coal pillar is an effective measure to ensure the stability of the roadway [35]. The grouting at the coal pillar side of the roadway wall reduced the deformation of the solid coal side, mainly because the grouting improves the bearing capacity of the coal pillar to the overburden. Additionally, the coal pillar side can bear more pressure transmitted by the overburden rock layer and share the stress of the solid coal side. The improved integrity of the coal pillar side also improved the ability to resist the deformation and damage of the roadway under the influence of mining. The deformation of the roadway decreased and the performance of coal pillar side was more obvious. Comparing the effect of the anchor cable support and grouting in the roadway,

we can see that the effect of grouting was better than anchor cable support. Comparing the effect of different grouting depths, it was found that the deformation reduction on the coal pillar side was not obvious as the grouting depth increased, and the difference in deformation between 2 m and 5 m after grouting was less than 0.05 m. Considering economic and other factors, therefore, 2 m grouting can be considered for the coal pillar side in the reinforcement process.

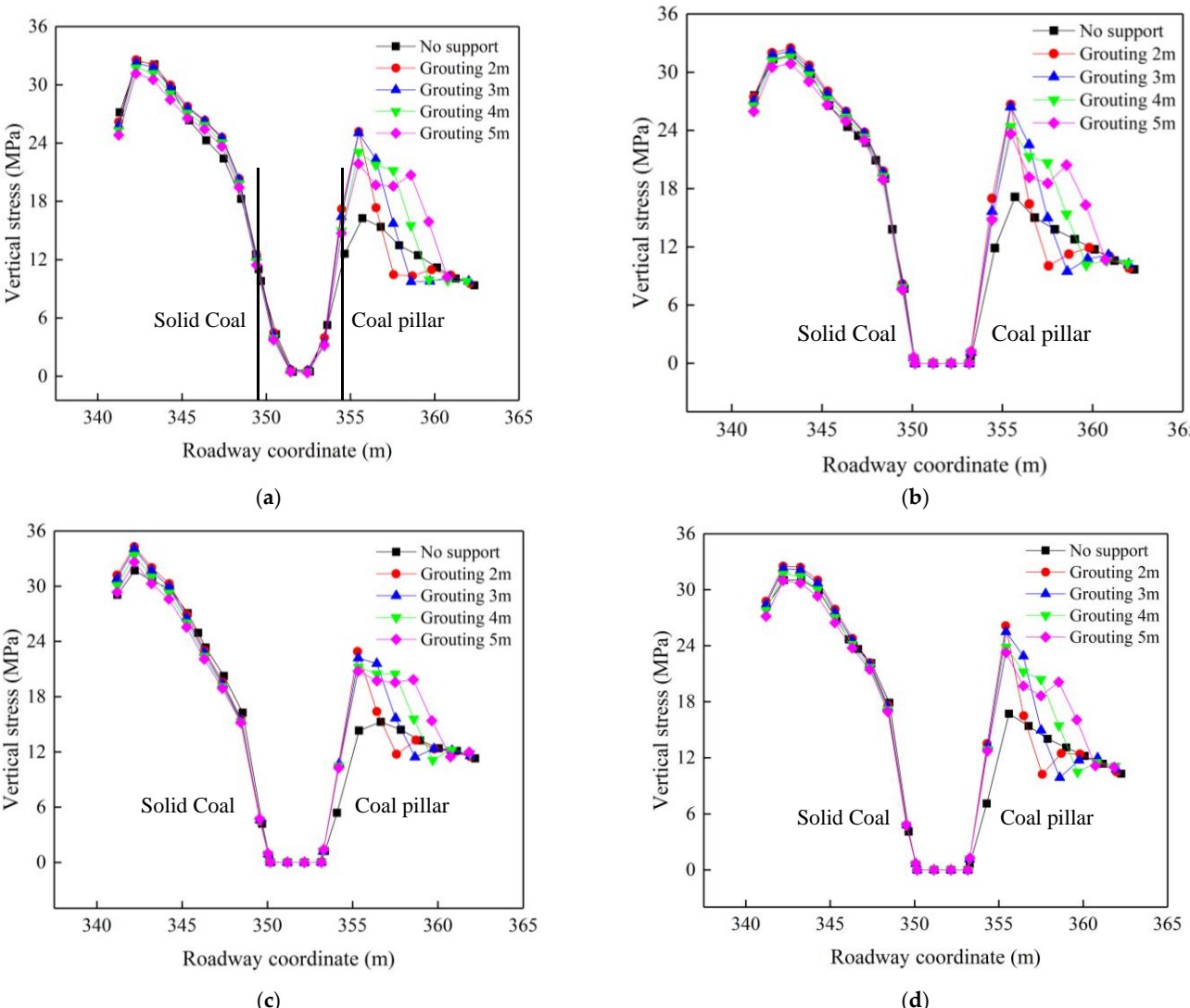

**Figure 12.** Vertical stress distribution in the two sides of the roadway at different grouting depths. (**a**) Monitoring Line A. (**b**) Monitoring Line B. (**c**) Monitoring Line C. (**d**) Monitoring Line D.

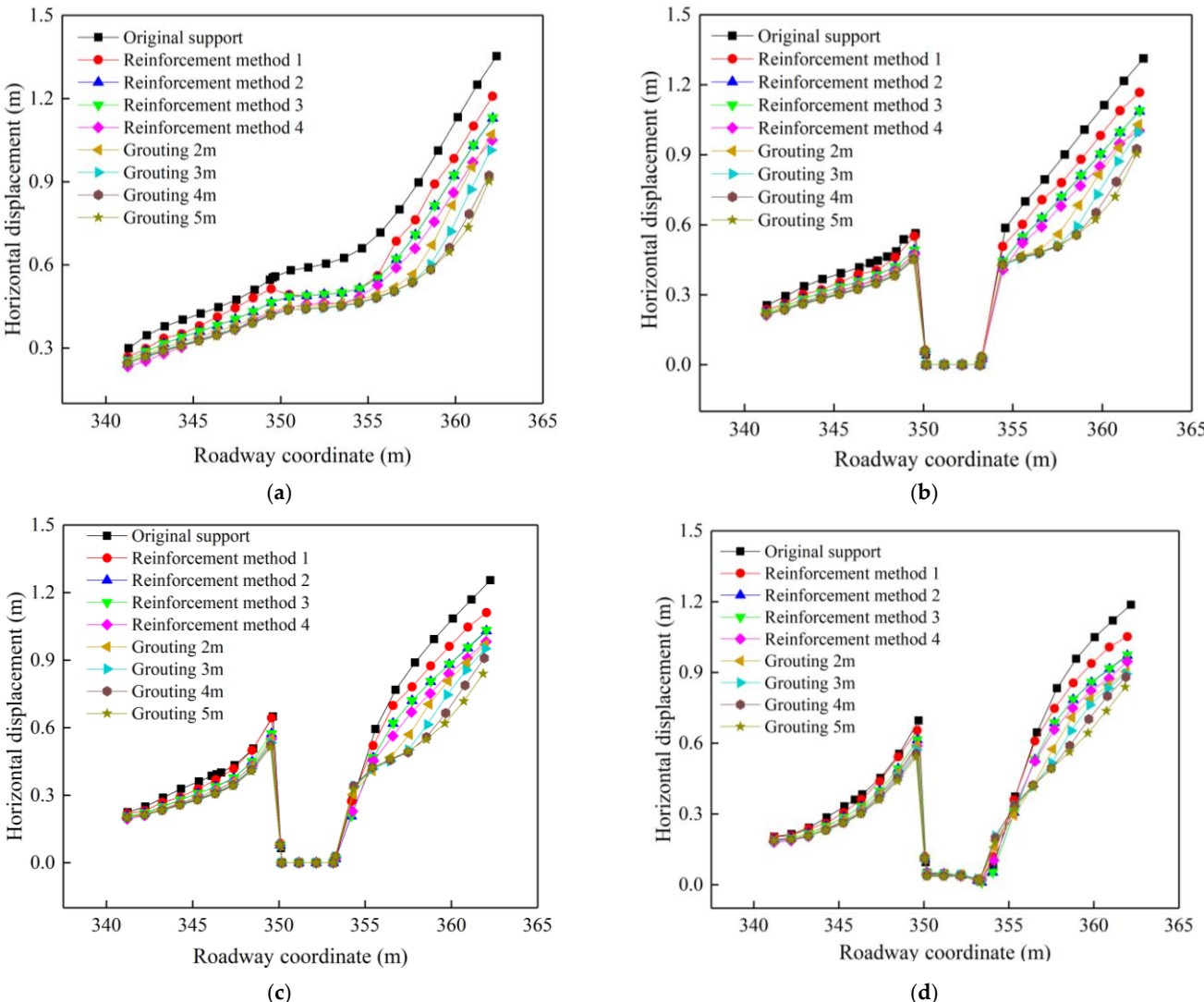

**Figure 13.** Horizontal displacement of the two sides of the roadway under different reinforcement methods of the roadway. (**a**) Monitoring Line A. (**b**) Monitoring Line B. (**c**) Monitoring Line C. (**d**) Monitoring Line D.

### 3.3. Stress and Displacement Distribution Pattern of the Roadway Wall under the Floor Reinforcement

3.3.1. Optimization of the Roadway Floor Support Method

The floor reinforcement support of the roadway includes floor anchor cable support and grouting. The grouting method of the floor is the same as that of the coal pillar side. The reinforcement support of the floor anchor cable includes the following three methods. (1) Five 4.15 m anchor cables were used; the first and last two were 0.7 m from the roadway wall, and the inter-row distance was 0.9 m × 2.0 m; (2) four anchor cables were used to support the vertical downward direction 1 m from the roadway wall, and the inter-row distance was 1.0 m × 2.0 m; (3) five anchor cables were used to support the floor of the roadway, with the first and last distance from the two sides being 0.3 m and the inter-row distance being 1.2 m × 2.0 m, corresponding to the way the roof of the roadway was supported. The anchor cable reinforcement method is shown in Figure 14.

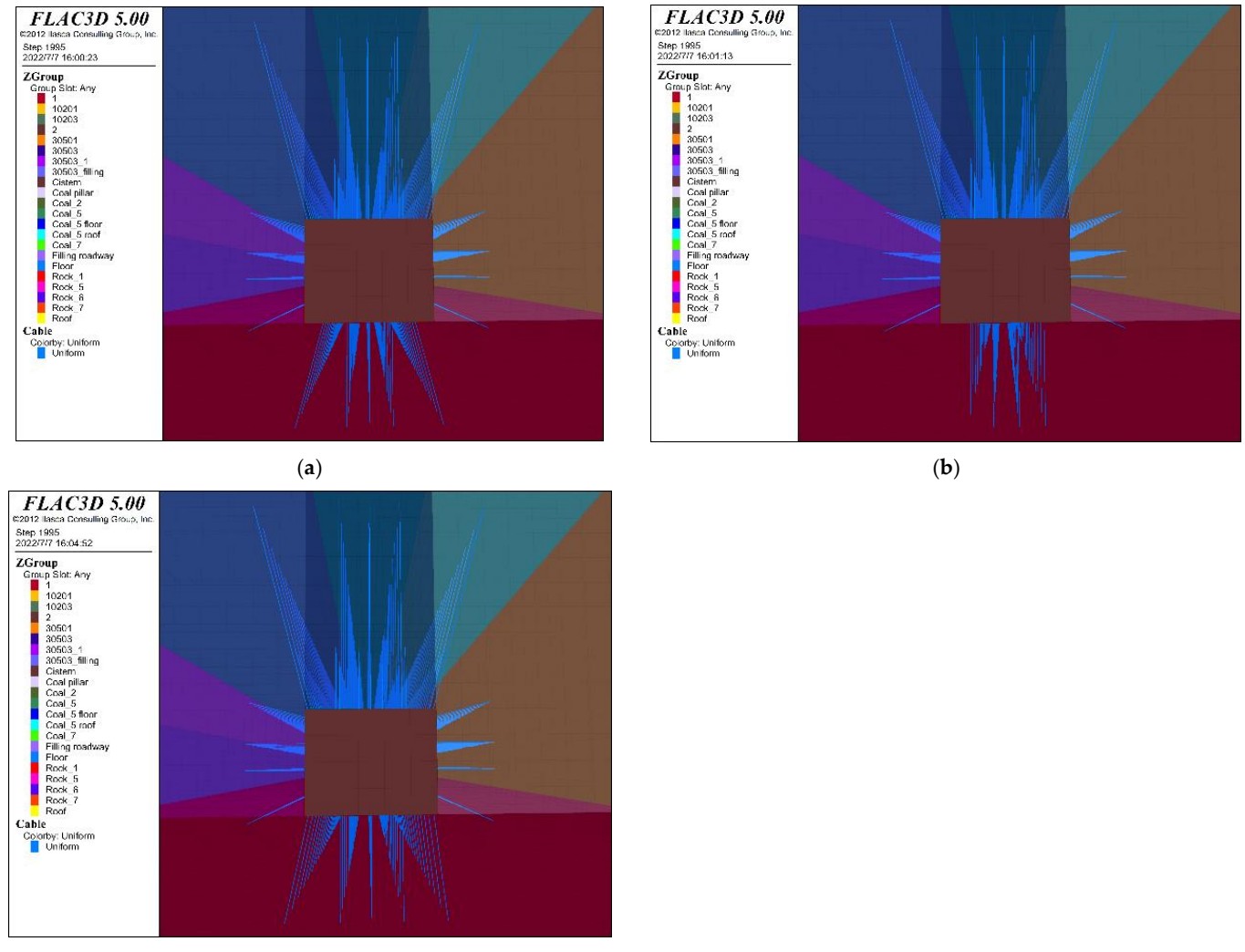

**Figure 14.** Anchor cable reinforcement method at the floor of the roadway. (**a**) Reinforcement method 1. (**b**) Reinforcement method 2. (**c**) Reinforcement method 3.

### 3.3.2. Deformation and Damage Characteristics of the Roadway Floor

When the working face was mined to 100 m strike, the vertical deformation distribution of the roadway floor under different anchor cable and grouting reinforcement methods is shown in Figure 15. Under the action of high stress on two sides of the roadway, the floor of the roadway had a continuous floor heave, the deformation damage of the floor was not reduced by increasing the anchor cable only, and the use of floor grouting obviously reduced the amount of floor heave of the roadway. The floor heave in the middle of the roadway floor under unsupported conditions was close to 0.5 m, and the anchor cable support was basically unchanged, indicating that the floor has been very broken under the influence of mining. The rock bearing capacity can not be improved by anchor cable support alone, and the maximum value of floor heave after grouting is 0.2 m, which is 60–70% lower than that before grouting. At 2 m below the floor of the roadway, the maximum value of floor deformation under the unsupported condition was 0.38 m, which decreased to 0.06 m after grouting. After the floor grouting, the sinking amount of the two sides of the roadway is also significantly reduced. This is because after the floor bearing capacity is improved, the force of the two sides cannot make the roadway more deform more, resulting in heaving and the coal body of the two sides will not sink.

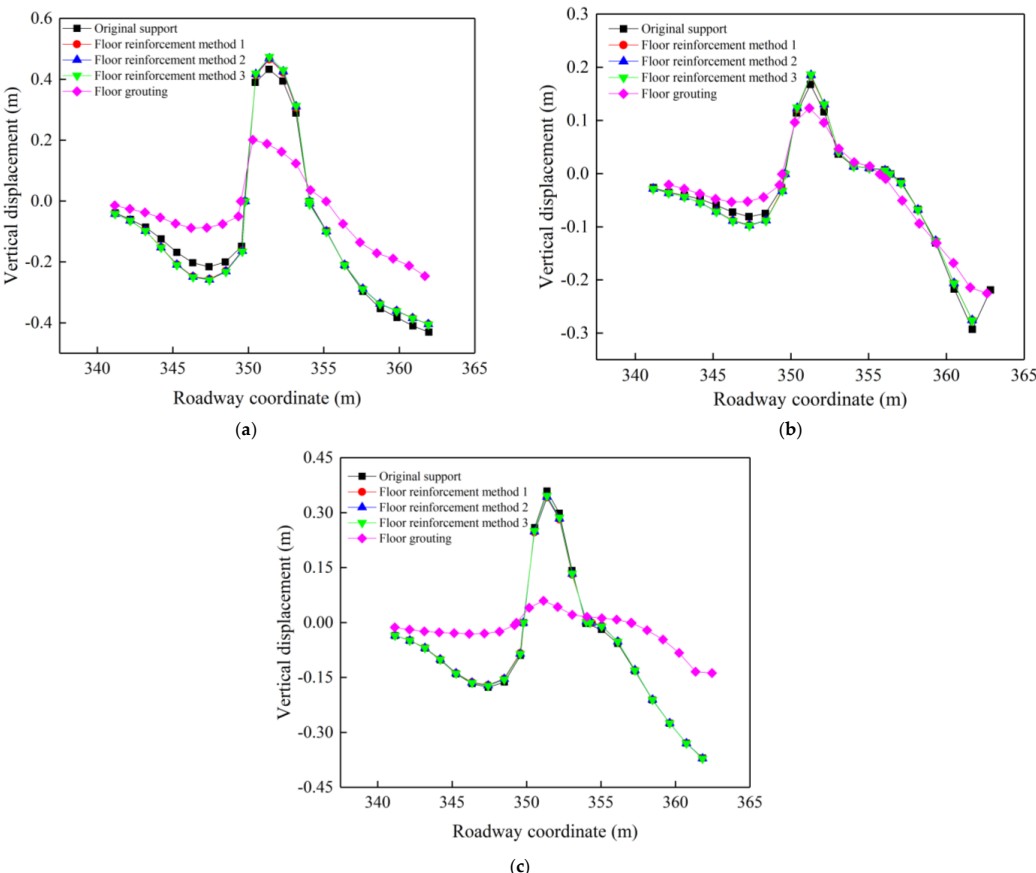

**Figure 15.** Vertical displacement of the floor under the floor reinforcement method of the roadway. (**a**) Floor. (**b**) 1 m below the floor. (**c**) 2 m below the floor.

In order to determine the optimal integrated reinforcement method for the roadway, a comprehensive analysis of the above reinforcement methods was performed. The deformation of the two sides of the roadway under different reinforcement methods are shown in Figures 16 and 17. The support effect of the anchor cable + the grouting support method was better than that of the simple anchor cable reinforcement support, but there was no obvious difference with the grouting reinforcement method. The comprehensive reinforcement method of the anchor cable + coal pillar side grouting + floor grouting at the side of the roadway had the best effect, with the lowest horizontal and vertical deformation, the reduction of vertical deformation was especially more obvious. The increase of vertical stress caused by mining is the direct cause of roadway deformation and damage. The floor heave of the roadway will further aggravate the increase of deformation on two sides. Based on this, it can be seen that when the strength of the floor is increased, on the one hand, it can bear more vertical stress caused by mining, so that the vertical stress can be transferred to a deeper depth. On the other hand, it can resist a larger floor heave in the floor.

The deformation of the roadway floor under different reinforcement methods is shown in Figure 18. From Figure 18, it can be seen that the use of floor grouting was still the most effective measure to control the floor heave. This is because the anchor cable and grouting reinforcement support increased the bearing capacity of the roadway wall. The stress concentration of the roadway wall increased, and the greater vertical stress was transferred to the floor, which increased the floor heave of the roadway.

Comparing only grouting and joint reinforcement support of the roadway floor, it can be seen that the deformation of the floor heave at depth increased to about 0.1 m after supporting the two sides, while the shallow part was basically unchanged. It can be seen that the joint anchor cable + grouting was the most effective measure to ensure the safety of

the roadway, as the deformation and damage of the roadway wall and floor under different reinforcement methods were combined.

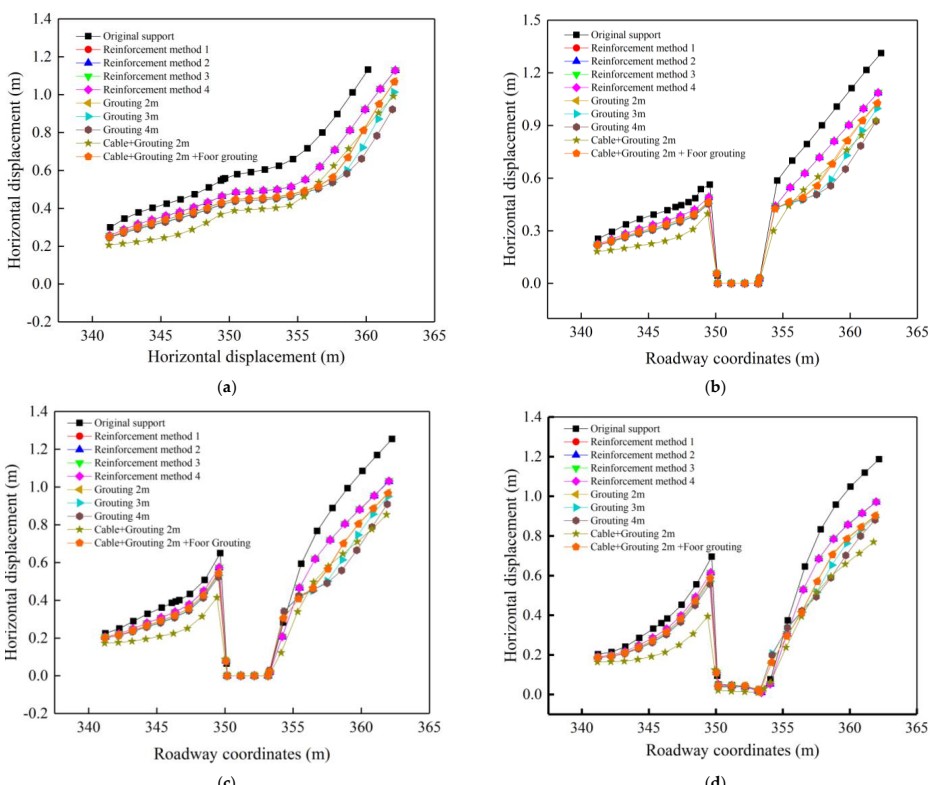

**Figure 16.** Horizontal displacement of the two sides of the roadway under different reinforcement methods. (**a**) Monitoring Line A. (**b**) Monitoring Line B. (**c**) Monitoring Line C. (**d**) Monitoring Line D.

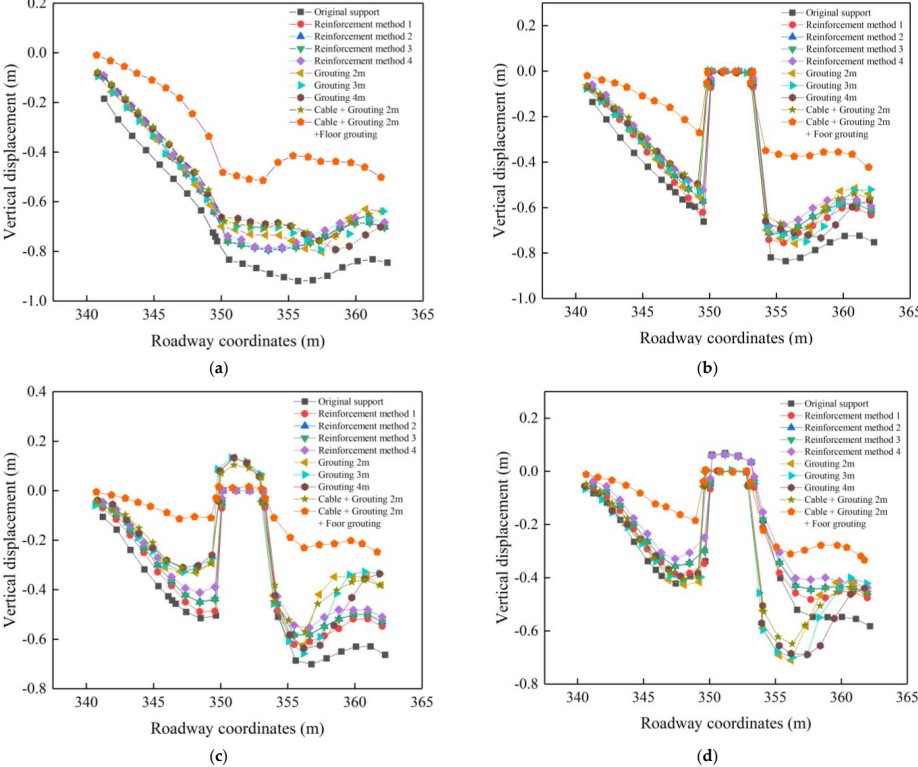

**Figure 17.** Vertical displacement of the two sides of the roadway under different reinforcement methods. (**a**) Monitoring Line A. (**b**) Monitoring Line B. (**c**) Monitoring Line C. (**d**) Monitoring Line D.

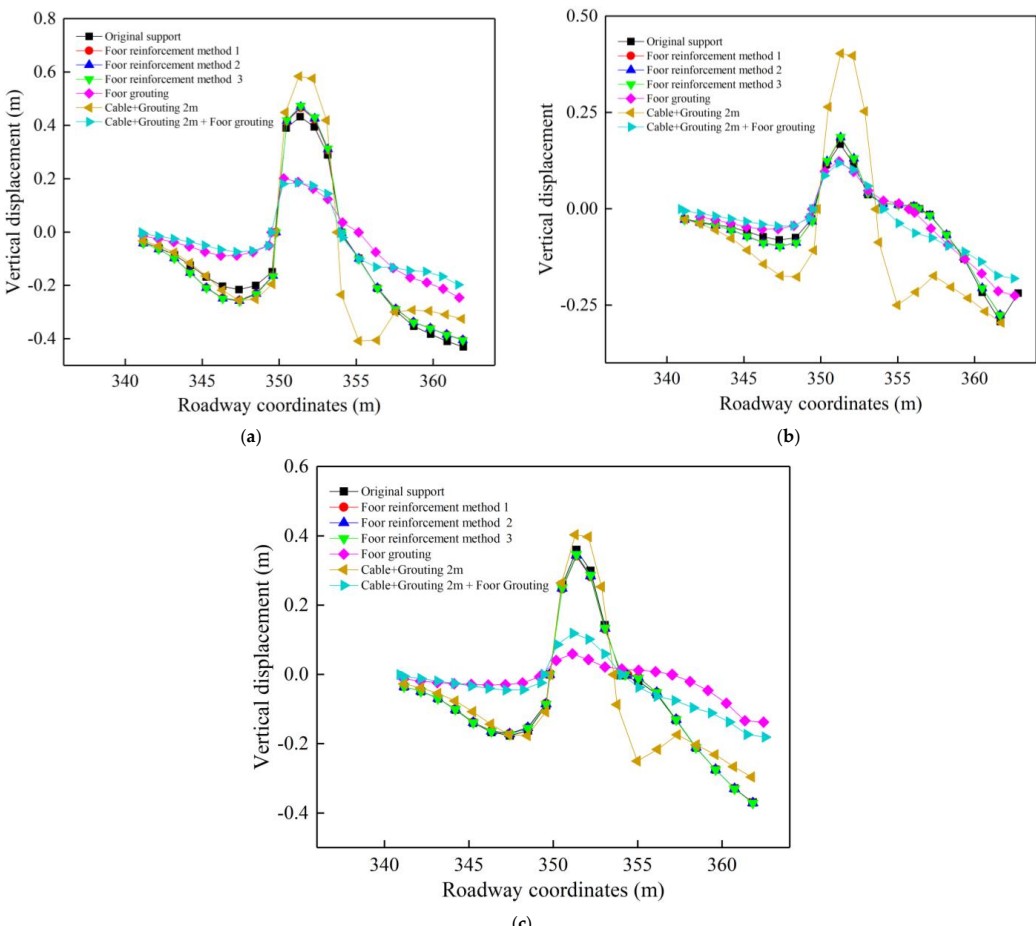

**Figure 18.** Vertical displacement of the floor under different floor reinforcement methods in the roadway. (**a**) Floor. (**b**) 1 m below the floor. (**c**) 2 m below the floor.

### 3.4. Field Application and Validation

In order to determine the effect of the joint anchor cable + grouting joint reinforcement support method to reduce roadway deformation and damage after application in the field, a laser rangefinder was used to monitor the roadway deformation within 250 m of the roadway ahead of the working face during the back mining of the 30,503 working face. The displacements of the roof and floor of the roadway and the two sides are shown in Figure 19. From Figure 19, it can be seen that affected by mining, the rapid deformation stage of the roadway was mainly concentrated within 50 m ahead of the working face, and the roadway deformation was large.

Near the working face, the deformation of the two sides was still about 600 mm, and the distortion of the ceiling and floor was still almost 350 mm. However, the deformation gradually decreased and stabilized with the increase of the distance ahead of the working face. This shows that after using the anchor cable + grouting support reinforcement, the deformation pattern of each monitoring point of the roadway was kept consistent and the deformation was better controlled. Compared with the deformation data of two sides of the roadway and the roof and floor without reinforcement support (Figure 20), the overall deformation of two sides was reduced by 50% and the roof and floor were reduced by about 40%. In the field observation, it is found that after the combined support of anchor cable and grouting, the floor heave and roadway wall heave still exists, but compared with the previous support, they are better controlled, which means that the existing support methods can better ensure the stability of the roadway during the recovery period.

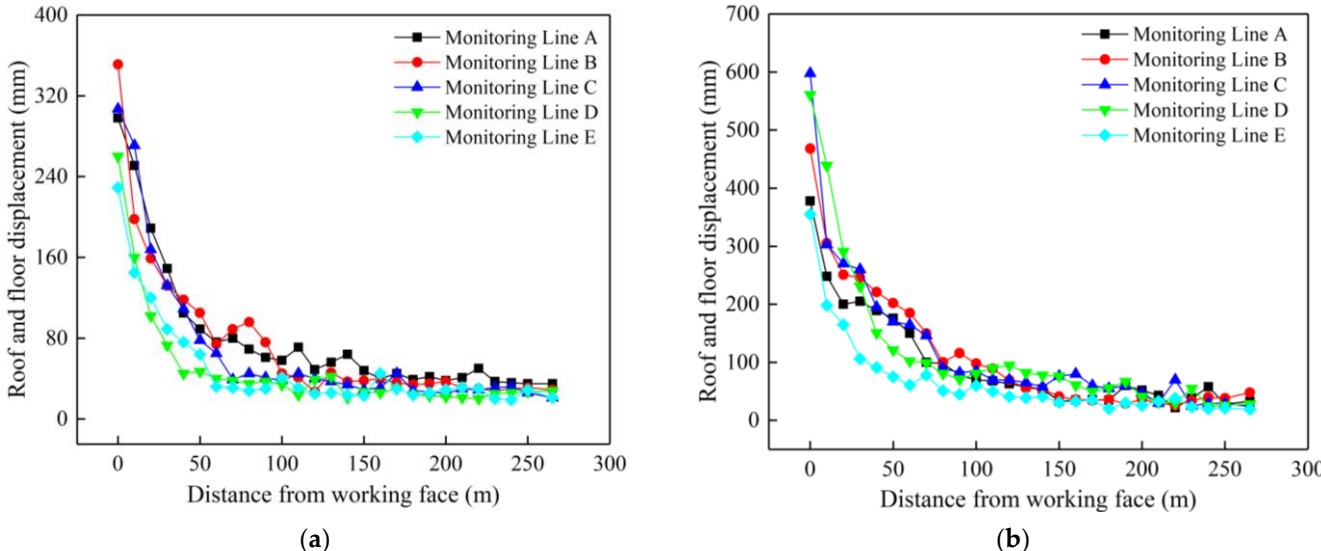

**Figure 19.** Deformation data after roadway reinforcement. (**a**) Roof–floor displacement. (**b**) Two sides displacement.

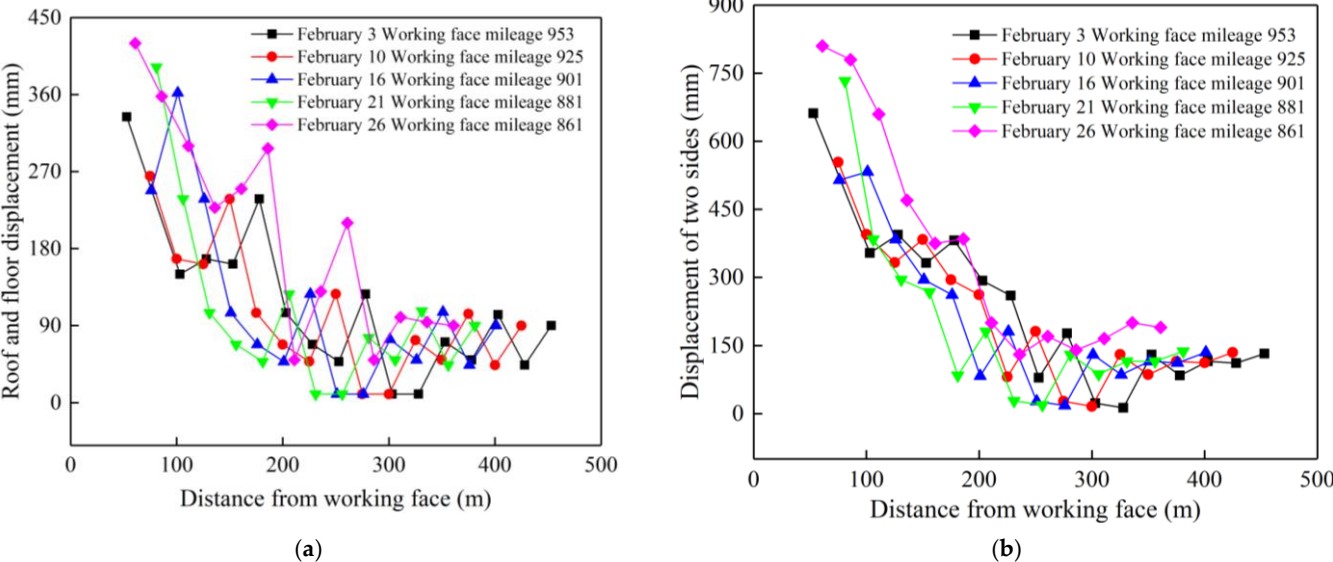

**Figure 20.** Deformation of the gob-side roadway wall. (**a**) Deformation of roof and floor. (**b**) Deformation of two sides.

## 4. Conclusions

### 4.1. Main Contributions

The close extra-thick coal seams are subject to the broken overburden of mined coal seams, and the deformation and damage of the roadway is serious, which affects the safe operation of the mine. In this paper, we studied the deformation and damage law of a gob-side roadway in close extra-thick coal seam by numerical simulation and field monitoring and compared and analyzed the deformation and damage characteristics of the roadway under different reinforcement support methods. Finally, the optimal reinforcement support method was determined and we verified it on site. The main contributions of the study are listed below. The gob-side roadway in close extra-thick coal seam has asymmetric deformation damage characteristics; the coal pillar side deformation was greater than that of the solid coal side. The large deformation of the coal body in the deep part of the roadway wall is an important reason for the continuous appearance of roadway wall heave in the coal pillar.

The horizontal and vertical displacement of the coal pillar side under various support methods was significantly different than that of the solid coal side. Each reinforcement support method reduces the deformation of the two sides of the roadway. Among them, increasing the length of anchor cable and arranging it in a fan shape can improve the support effect remarkably. The bearing capacity and stability can be greatly increased by grouting at the coal pillar side. Grouting to strengthen the integrity of the floor is the most effective way to avoid and control floor heave since its effect on regulating floor heave is significantly better than that of anchor cables. Therefore, the comprehensive reinforcement method of anchor cable + coal pillar side grouting + floor grouting at the roadway wall has the best effect, especially the vertical deformation.

### 4.2. Limitations and Prospects

The limitation of this paper is that the optimized support results for the gob-side roadway in the close extra-thick coal seam obtained in this study are based on the horizontal coal seam, but whether it is applicable to other inclined coal seams needs further research and verification. Additionally, based on the understanding of the limitations of this paper, in the future, the author will conduct large-scale physical similarity simulation to further study the instability characteristics of the gob-side roadway in close extra-thick coal seam, and provide guidance for roadway instability control under this condition.

**Author Contributions:** B.Z.: Writing—original draft, investigation, conceptualization. S.H.: Validation, conceptualization, methodology, funding acquisition. X.H.: Validation, writing—review and editing. L.G.: Methodology, conceptualization. Z.L.: Investigation, formal analysis. D.S.: Writing—review and editing, formal analysis. F.S.: Methodology, formal analysis. All authors have read and agreed to the published version of the manuscript.

**Funding:** This work was financially supported by the Postdoctoral Research Foundation of China (Project No. 2021M700371). We thank anonymous reviewers for their comments and suggestions to improve the manuscripts.

**Institutional Review Board Statement:** Not applicable.

**Informed Consent Statement:** Not applicable.

**Data Availability Statement:** Not applicable.

**Acknowledgments:** The authors confirm that there are no conflicts of interest associated with this publication. The authors gratefully thank the anonymous reviewers for their constructive comments on improving the presentation. All authors have agreed to the listing of authors.

**Conflicts of Interest:** The authors declare no conflict interests.

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
