# Peer review of "Research on Deformation and Failure Control Technology of a Gob-Side Roadway in Close Extra-Thick Coal Seams"

_sustainability, doi:10.3390/su141811246_

Round 1

Reviewer 1 Report

The serious deformation and failure of gob-side roadway has been a difficult problem that perplexes the mine safety production. The presented work addresses a very interesting work related to the research on deformation and failure control technology of gob-side roadway in close extra-thick coal seam. Please consider some minor comments to improve the manuscript.

1.I suggested to change the 30503 transport roadway into goaf in Fig.1.

2. It would be better to mark the model size in Fig.4.

3. The first letters of words in Table 2 shall be capitalized.

4. The main limitations of the study should be highlighted in the conclusion section.

5.The reasons why different support schemes have different support effects need to be further explained.

6. English needs to be further improved.

7. Conclusions should be streamlined.

8. Several references should be added, e.g., Feng, Q., Jin, J., Zhang, S. et al. Study on a Damage Model and Uniaxial Compression Simulation Method of Frozen–Thawed Rock. Rock Mech Rock Eng 55, 187–211 (2022). https://doi.org/10.1007/s00603-021-02645-2; Li, Wenshuai, Jiang, Bangyou, Gu, Shitan, Yang, Xuxu, Faiz U.A. Shaikh. Experimental study on the shear behaviour of grout-infilled specimens andmicromechanical properties of grout-rock interface. Journal of Central South University(2022)29: 1686-1700. https://doi.org/10.1007/s11771-022-5026-5

Reviewer 2 Report

Mathematics (ISSN 2071-1050)

Manuscript ID: sustainability-1854808

Reviewer Comments

Paper title: Research on deformation and failure control technology of gob-side roadway in close extra-thick coal seam

The present manuscript describes the deformation and damage law of the close extra-thick coal seam gob-side roadway through numerical simulation, compared and analyzed the deformation and damage characteristics of the roadway under different reinforcement support methods, determined the optimal reinforcement support method, and carried out field verification.

A manuscript has a practical application and also provides important theoretical for the next studies.

The paper can be accepted for publication after providing the corrections mentioned below.

Point 1. Point 1. The abstract section sounds unclear. It is too large in volume. You must present it short: The abstract should follow the MDPI style of structured abstracts:

- Background (place the question addressed in a broad context and highlight the purpose of the study);

- Methods (describe briefly the main methods);

- Results (summarize the article's main findings);

- Conclusion (indicate the main conclusions or interpretations).

Point 2. Keywords need to be modified. Please use words not combinations of words or phrases.

Point 3. In the Introduction section, an enhanced literature review is required. For this study, the authors have used only 29 literature sources. It seems insufficient for such type of research. It will be great if the authors show some description in context – Why it is important to conduct this study?

Point 4. The aim and the tasks must be highlighted at the end of the Introduction section.

Point 5. Figure 3 is unclear. It will be better to show it as a schematic diagram.

Point 6. Why FLAC3D was selected as the tool of numerical modelling. Provide your interpretation.

Point 7. Please add numeration of the Rockiness of the Table 1.

Point 8. Text from the section 3.1 “The monitoring line A is located at the height of the roof of the roadway. With an interval of 1 m, a total of 4 monitoring lines are arranged. The floor monitoring line starts from the position of 1 m from the floor, with an interval of 1 m, and a total of 4 lines are arranged.” Must be placed in the Section 2 as well as Figure 5 (30503 gob-side roadway monitoring line layout). Moreover, it must be described in more details for better understanding.

Point 9. The novelty of the paper must be highlighted in the conclusions section.

Point 10. Please provide a short description of further research.

Point 11. Do not use numeration of paragraph in the Conclusion section.

Point 12. It is quite difficult to read the paper. Why do authors not prepare the paper using a commonly known IMRaD structure?

The study should follow a conventional pattern. Methods, Results and Discussion need to be mentioned clearly so that the readers can easily understand the gist of the manuscript.

Point 13. Please consider the suggested research in your paper when enhancing the literature review (enhanced review of non-China authors only are more than welcome). I believe they are worth considering in your paper.

I. Slashchov, I., Shevchenko, V., Kurinnyi, V., Slashchova, O., & Yalanskyi, O. (2019). Forecast of potentially dangerous rock pressure manifestations in the mine roadways by using information technology and radiometric control methods. Mining of Mineral Deposits, 13(4), 9-17. https://doi.org/10.33271/mining13.04.009

II. Małkowski, P., Niedbalski, Z., Majcherczyk, T., & Bednarek, Ł. (2020). Underground monitoring as the best way of roadways support design validation in a long time period. Mining of Mineral Deposits, 14(3), 1-14. https://doi.org/10.33271/mining14.03.001

III. Dychkovskyi, R., Shavarskyi, Ia., Saik, P., Lozynskyi, V., Falshtynskyi, V., & Cabana, E. (2020). Research into stress-strain state of the rock mass condition in the process of the operation of double-unit longwalls. Mining of Mineral Deposits, 14(2), 85-94. https://doi.org/10.33271/mining14.02.085

IV. Begalinov, A., Almenov, T., Zhanakova, R., & Bektur, B. (2020). Analysis of the stress deformed state of rocks around the haulage roadway of the Beskempir field (Kazakhstan). Mining of Mineral Deposits, 14(3), 28-36. https://doi.org/10.33271/mining14.03.028

Point 14. In general, I must admit that a very good study was performed, and I will recommend your paper for publication after careful revision.

Round 2

Reviewer 2 Report

Dear Dr Shengquan He,

I am more than satisfied with the corrections provided by you.

This study is an important contribution to sustainable mining technologies.

Congratulations to the authors.